# Retrospective analysis of malaria prevalence over ten years (2015–2024) at Bichena Primary Hospital, Amhara Region, Ethiopia

Awoke Minwuyelet[1]*, Delenasaw Yewhalaw[2,3], Getnet Atenafu[1]

1 Department of Biology, College of Natural and Computational Science, Debre Markos University, Debre Markos, Amhara, Ethiopia, 2 Tropical and Infectious Diseases Research Center, Jimma University, Jimma, Ethiopia, 3 School of Medical Laboratory Sciences, Faculty of Health Sciences, Jimma University, Jimma, Ethiopia

* awokeminwuyelet5@gmail.com

## Abstract

### Background

Malaria remains a significant public health challenge in Ethiopia, hindering the country's productivity and development. While malaria incidence had decreased by 2018, and Ethiopia is working towards eliminating the disease by 2030, outbreaks still occur even in areas of low endemicity. Therefore the aim of this study was to assess the ten-year trend in malaria prevalence from 2015 to 2024 at Bichena Primary Hospital in the Amhara region of northwestern Ethiopia.

### Materials and methods

A retrospective review of malaria blood film examination results was conducted using laboratory registration logbooks at Bichena Primary Hospital. Data collection was carried out from December 30, to January 14. The data were collected using a data collection sheet and entered into the Statistical Package for Social Sciences (SPSS) version for analysis. Bi-variable and multi-variable regression analyses and Pearson's chi-square test were used to examine associations and differences in malaria prevalence trends across factors such as sex, age, year, and season and *Plasmodium* species. Descriptive statistics were also used to summarize the sociodemographic characteristics of the study participants and the results were presented in graphs, tables and texts.

### Results

Out of the 24,107 malaria blood films examined, 4,322 (17.9%, 95% CI: 17.4%-18.4%) tested positive for *Plasmodium* infections. Of the confirmed cases, 58.7% were *P. vivax,* 28.6% were *P. falciparum,* and 12.2% were mixed infections. *P. vivax* was the predominant species throughout the study period (2015-2024), except for the years 2016 and 2018, when *P. falciparum* was more prevalent. Subsequently, an increase in malaria cases was reported, with the highest proportion recorded in 2024 (26.8%)

**Data availability statement:** All relevant data are within the article and its Supporting information files.

**Funding:** The author(s) received no specific funding for this work.

**Competing interests:** The authors have declared that no competing interests exist.

**Abbreviations:** ITNs, Insecticide-treated nets; IRS, Indoor residual spraying; LLINs, long-lasting insecticidal nets; SPSS, Statistical Package for Social Sciences; WHO, World Health Organization.

and the lowest in 2018 (4%). The likelihood of malaria prevalence was 1.28 times higher in males than in females. Additionally, the chance of malaria prevalence was 1.27 times higher in the 15-24 age group compared to other age groups. The study revealed a significant rise in malaria prevalence, highlighting that malaria remains a major public health issue in the study area. There were also pronounced seasonal variations in malaria cases each year, with males and younger adults being more affected than females, older age groups, and children under five.

## Conclusions and recommendations

Malaria prevention and control efforts need to be strengthened, focusing on regional differences. Ongoing research on diagnostic challenges, parasite elimination, and mosquito infectivity after malaria treatment is essential.

### Introduction

Malaria continued to pose a significant public health challenge, particularly in tropical and subtropical regions. In 2023, approximately 263 million malaria cases and 597,000 deaths were reported worldwide, result in in a mortality rate of 13.7 per 100,000. This represented an increase of 11 million cases compared to the previous year, with incidence raising to 58.6 cases per 1,000 of the at-risk population in 2022. According to the World Health Organization (WHO) in its 2024 malaria report, the African region remained the most severely impacted by malaria in 2023, accounting for an estimated 94% of global cases. Most cases and deaths recorded in sub-Saharan Africa, where the disease burden was most pronounced [1].

From 2020 to 2024, malaria cases and deaths increased globally [2]. The distribution and epidemiology of malaria have become increasingly complex, with notable variations in prevalence, species distribution, and significant challenges in control efforts across different regions. While some countries have made considerable progress in reducing cases, others continue to struggle, particularly in areas with high forest cover, heavy rainfall, and disadvantaged populations [3,4]. Children under five, who account for 76% of global malaria deaths, are especially vulnerable due to their developing immune systems and reduced maternal immunity [5]. Achieving malaria elimination will require addressing asymptomatic infections, particularly in school-age children, and ensuring protection for the most at-risk groups [5,6].

Malaria remains a critical public health concern in Ethiopia, despite significant progress in control measures over the past two decades. The country has experienced varying levels of malaria prevalence, with key challenges stemming from environmental factors, vector resistance, and socio-political dynamics [7–10]. Between 2013 and 2018, Ethiopia performed significant decline in malaria cases, attributed to effective interventions such as insecticide-treated nets (ITNs) and indoor residual spraying (IRS) [11,12]. Similarly, malaria cases have decreased from 3.5 million to fewer than one million by 2019, and malaria-related deaths fell from 3,000 in 2010–212 in 2021 [13]. The Ethiopian government has set ambitious goals for malaria elimination by 2030, supported by evidence-based programs and increased funding for health initiatives [14,15]. However,

the Ethiopian Federal Ministry of Health (FMOH) annual performance report for 2022 revealed a troubling 66% increase in malaria cases compared to 2019, with the number of cases rising from 904,495 in 2019–1,504,405 in 2022 [13]. According to the WHO report, from January 1 to October 20, 2024, Ethiopia recorded over 7.3 million malaria cases and 1,157 deaths, resulting in a case fatality rate of 0.02%.

Malaria in Ethiopia is not uniformly distributed across the country. For example, the Amhara Region accounts for 31% of the national malaria burden [15]. Although overall prevalence has decreased, there has been a rise in unpredictable outbreaks in areas previously regarded as "malaria-free" [15]. This shift in transmission patterns presents new challenges for malaria control efforts. Recent evidence indicates that the risk of relapsing *Plasmodium vivax* malaria is elevated following an acute *P. falciparum* infection [8,16–18]. Furthermore, Ethiopia's malaria control programs are struggling to meet elimination targets due to the persistent burden of recurrent *P. vivax* infections [18].

In some regions, malaria incidence has declined due to effective interventions such as ITNs, IRS, and antimalarial medications [18]. Despite widespread coverage of control measures like long-lasting insecticidal nets (LLINs) and IRS, certain areas continue to experience unexpectedly high malaria prevalence [18,19]. Progress in malaria control has slowed in recent years due to factors like drug resistance, insecticide resistance, the introduction of new species, such as *Anopheles stephensi,* which complicate control efforts, and inadequate funding for malaria programs [1,12–14]. In addition, climate change and environmental anomalies have been linked to increased malaria transmission, particularly in previously low-risk areas [15].

Despite the progress made, the resurgence of malaria and the emergence of new challenges underscore the need for adaptive strategies and enhanced surveillance to sustain control efforts. This study aimed to assess the trends in malaria from 2015 to 2024 at Bichena Primary Hospital in the Amhara Region, Ethiopia.

## Materials and methods

### Study area

The study was conducted at Bichena Primary Hospital from 2015 to 2024, using the hospital's laboratory logbook. The hospital is located in the town of Bichena in Enemay district, about 263 kilometers from Addis Ababa. The town is located in the East Gojjam Zone of the Amhara Region on a hill above the Abay River. The geographical coordinates of the town are 10°27'24 "N and 38°12'23 "E with an altitude of 2,541 meters above sea level. The tropical climate of the region, especially during the rainy season, favors the transmission of malaria. The hospital provides comprehensive medical care, including malaria treatment, not only to the local community but also to residents of the neighboring districts of Enemay, Shebel Berenta, Debay Tilat Gin and Enarg Enawuga. Malaria, a major public health problem, is frequently diagnosed by the hospital's laboratory staff. Recently, there has been a malaria outbreak in the region characterized by a significant incidence of *P. falciparum* and *P. vivax* since June 2024. This makes the hospital an important site for assessing the malaria trend.

### Study design and period

The study utilized a retrospective study of malaria prevalence over a ten-year period by reviewing the records of malaria blood film examinations in the laboratory registration logbook of Bichena Primary Hospital from 2015 to 2024. Data collection was performed from December 30, 2024, to January 14, 2025. All registered malaria blood films were included in the study except those with incomplete data or illegible records.

### Study population, sample size and sampling technique

The study population consisted of all cases of suspected malaria who requested testing for the parasite in the laboratory department of Bichena Primary Hospital from 2015 to 2024. This included all individuals who underwent malaria blood film testing and were recorded in the laboratory logbooks.

The sample size included all malaria blood film tests documented in the laboratory logbooks during the specified ten-year period (a total of 24,143 cases). However, malaria blood film examinations with incomplete data or illegible records (36 participants) were excluded from the study.

 

## Data collection and analysis methods

**Data collection procedures.** The data source consisted of detailed records from the hospital's laboratory logbooks, documenting the period from January 1, 2015, to December 31, 2024. The data was collected by the first author and other trained laboratory personnel using a data extraction sheet (paper form) established for this study. The data extraction sheet included information on the blood film results (either positive or negative), patient demographics (such as gender and age), the date of diagnosis (including month, season, and year), and, for positive cases, the species identified (*P. falciparum, P. vivax,* and their mixed species).

***Sample collection and laboratory diagnosis of malaria***: After taking the patient's history, the physician sent them to the laboratory department with a request for a blood film. Laboratory personnel aseptically collected either a capillary or venous blood sample and prepared both thin and thick blood films. These blood films were stained with a 10% fresh Giemsa stain solution for 10 minutes. Once stained, the slides were air-dried, and both thick and thin blood films were examined under a magnification of 100× to detect and identify *Plasmodium* parasites, including both asexual and gametocyte stages. A slide was considered negative if no *Plasmodium* parasites were observed after examining 200 fields. Blood smears were verified by at least two laboratory technicians before the results were reported. In cases of discordant results, a third senior technologist reexamined the slide, and the final result was based on their report.

**Data analysis.** After data collection using data extraction sheet, the data were entered into the SPSS version 25.0 software [20] and checked for completeness and accuracy prior to analysis. Descriptive statistics, including frequencies and percentages, were used to summarize annual malaria cases, as well as cases categorized by species (*P. falciparum, P. vivax* and their mixed). Trend analysis was conducted to examine changes in malaria prevalence over time, determining whether the prevalence increased, decreased, or remained stable from 2015 to 2024. Seasonal variation in malaria cases was also analyzed, with data correlated to climate patterns (wet and dry seasons) to identify transmission peaks. The data were further aggregated by age groups and gender to assess the distribution of malaria cases across demographic categories. The study results were presented in figures, tables and texts.

Statistical analyses, including bivariate and multivariable regression as well as the Pearson chi-square test, were applied to identify significant differences in malaria prevalence over the study period, with a significance level of $P < 0.05$ considered statistically significant.

## Ethical approval and consent to participate

All procedures were conducted following the relevant guidelines and regulations. Debre Markos University has an established Ethical Review Committee for PhD studies to approve research involving human participants at the University level. Ethical clearance for this study was granted on December 30, 2024, with reference number DMU/RTTD/75/10/2024, by the Institutional Research Ethics Review Committee of the Research and Technology Transfer Directorate of Debre Markos University. After explaining the study's objectives and methodology, verbal consent was obtained from the chief executive officer of Bichena Primary Hospital before data collection. Since the study used secondary data from the registration logbook, the Institutional Research Ethics Review Committee of Debre Markos University waived informed consent from individual participants. Patient confidentiality was strictly maintained throughout the data collection and analysis process.

## Results

### Demographic characteristics and malaria prevalence

Of the 24,143 malaria blood films examined from January 1/2015 to December 31/2024, 24,107 were included in the final analysis. The majority of participants 12,873(53.4%) were male. Ages ranged from 1 day to 103 years, with a mean

of 27.2 years (±17.5 SD). Among age categories, the majority (22.1%) of blood films examined were from the age group 15–24 years old.

Of the 24,107 blood films examined microscopically for malaria diagnosis, 4,322 (17.9%; 95% CI: 17.4%–18.4%) were confirmed as malaria cases. Among these, 2,501 (19.4%) were male. The highest number of confirmed cases was observed in the 15–24 age group (1,346 cases, 22.3%), followed by the 35–44 age group (554 cases, 18.7%). The lowest prevalence was recorded in children under five years old, with 258 cases (8.9%) (Table 1).

## Malaria distribution

Despite the noticeable fluctuations in overall malaria trends over the ten-year period in the study area, malaria cases were consistently reported throughout the year. However, significant variations were observed between genders, age groups, and in monthly, seasonal, and yearly patterns.

The odds of malaria prevalence were 1.28 times higher in males than in females (95% CI: 1.20–1.37). Regarding age groups, the odds were 0.42 times lower in those under 5 years old (95% CI: 0.36–0.48), 0.84 times lower in the 5–14 years age group (95% CI: 0.73–0.96), and 1.27 times higher in the 15–24 years age group (95% CI: 1.15–1.40) compared to those aged 45 and above. Additionally, the odds of malaria prevalence were 0.87 times lower in autumn (95% CI: 0.77–0.98), 2.49 times higher in summer (95% CI: 2.28–2.72), and 1.32 times higher in spring (95% CI: 1.19–1.45) compared to the winter season (Table 1).

## Annual trends in malaria prevalence and proportions of *Plasmodium* species

The trend of microscopy-confirmed malaria cases, as indicated by blood film examinations, showed significant fluctuations over the years. The highest number of blood film examinations occurred in 2024, with a total of 10,054 suspected cases (41.7%), and this year also saw the highest number of confirmed malaria cases (26.9%), and followed by 2022 with 19.1% confirmed cases. The lowest number of suspected malaria cases was recorded in 2019, with a total of 758 cases, but with a higher malaria prevalence of 19%. The lowest confirmed malaria cases were observed in 2018, with a prevalence of 4% (Table 2 and Fig 1).

**Table 1. Sex, age, and seasonal variations in malaria prevalence and bi-variable and multi-variable analysis of associated factors among patients who requested malaria examinations at Bichena Primary Hospital from 2015 to 2024.**

| Variables | | Positive (%) | Negative (%) | Total (%) | OR (95%,CI) | | |
|---|---|---|---|---|---|---|---|
| | | | | | COR | AOR (95% CI) | *P* value |
| Sex | Male | 2,501(19.4) | 10,372 (80.6) | 12873(53.4) | 1.25(1.17-1.33) | 1.28(1.2-1.37) | 0.00 |
| | Female | 1,821 (16.2) | 9,413 (83.8) | 11234(46.6) | 1 | 1 | |
| Age category | <5 | 258 (8.9) | 2,636 (91.1) | 2894(12) | 0.44(0.38-0.51) | 0.42 (0.36-0.48) | 0.00 |
| | 5-14 | 422(15.8) | 2,2663 (83.2) | 2663(11) | 0.84(0.74-0.95) | 0.84(0.73-0.96) | 0.01 |
| | 15-24 | 1,346 (22.3) | 4,699 (77.7) | 6045((25.1) | 1.27(1.15-1.41) | 1.27(1.15-1.40) | 0.00 |
| | 25-34 | 967(18.2) | 4,351 (81.8) | 5318(22.1) | 0.99(0.89-1.09) | 1.01(0.91-1.12) | 0.85 |
| | 35-45 | 554(18.7) | 2,412 (81.3) | 2966(12.3) | 1.02(0.91-1.15) | 1.02(0.90-1.16) | 0.73 |
| | ≥45 | 775(18.4) | 3,446(816) | 4221(10) | 1 | 1 | |
| Seasons | Winter | 843(12.9) | 5,664 (87.1) | 6510(27) | 1 | 1 | |
| | Autumn | 466(11.7) | 3,503 (88.3) | 3969(16.5) | 0.89(0.79-1.01) | 0.87(0.77-0.98) | 0.03 |
| | Summer | 1,951(26.8) | 5325(73.2) | 7276(30.2) | 2.46(2.25-2.69) | 2.49(2.28-2.72) | 0.00 |
| | Spring | 1,062(16.7) | 5,290 (83.3) | 6352(26.3) | 1.35(1.22-1.49) | 1.32(1.19-1.45) | 0.00 |
| Total | | 4322(17.9) | 19785(82.1) | 24107 (100) | | | |

**AOR**: Adjusted odd ratio, **CI**: Confidence interval, **COR**: Crude odd ratio, **OR**: odd ratio.

**Table 2. Annual, seasonal, and monthly prevalence of malaria cases, positivity, and proportion of *Plasmodium* species among patients tested for malaria at Bichena Primary Hospital from 2015 to 2024.**

| Periods | Positivity rate n (%) | Negativity rate n (%) | χ² (P -Value) | *Plasmodium* species proportion (n=4322) | | | χ² (P Value) |
|---|---|---|---|---|---|---|---|
| **Years** | | | | *P.falciparum* (%) | *P.vivax*(%) | Mixed (%) | |
| 2015 | 192(10.3) | 1672(89.7) | 938.2 (0.00) | 88(45.8) | 100(52) | 4(2.1) | 172.1 (0.00) |
| 2016 | 370(9.7) | 3445(94.3) | | 258(69.7) | 98(26.4) | 14(3.7) | |
| 2017 | 107(5.6) | 1790(94.4) | | 32(29.9) | 63(58.9) | 12(11.2) | |
| 2018 | 34 (4) | 812(96) | | 18(52.9) | 14(41.2) | 2(5.9) | |
| 2019 | 144(19) | 614(81) | | 38(26.4) | 76(52.8) | 30(20.8) | |
| 2020. | 88(11) | 711(89) | | 34(38.6) | 50(56.8) | 4(4.5) | |
| 2021 | 118(12.4) | 834(87.6) | | 42(35.6) | 66(55.9) | 10(8.5) | |
| 2022 | 262(19.1) | 1,108(80.9) | | 108(41.2) | 150(57.2) | 4(1.5) | |
| 2023 | 306(17.5) | 1,1446(82.5) | | 70(22.9) | 206(67.3) | 30(9.8) | |
| 2024. | 2,701(26.9) | 7353(73.1) | | 772(28.6) | 1586(58.7) | 343(12.7) | |
| **Seasons** | | | | | | | |
| Winter | 843(12.9) | 5667(87.1) | 590.5 (0.00) | 265(31.4) | 511(60.6) | 67(7.9) | 21.6 (0.002) |
| Autumn | 466(11.7) | 3503(88.3) | | 176 (37.7) | 248(53.2) | 42(9) | |
| Summer | 1951(26.8) | 5325(73.2) | | 639(32.7) | 1096(56.2) | 216(11.1) | |
| Spring | 1062(16.7) | 5290(83.3) | | 380(35.8) | 554(52.2) | 128(12.1) | |
| **Months** | | | | | | | |
| January | 155(9.4) | 1491(90.6) | 943.1 (0.00) | 58(37.4) | 91(58.7) | 6(3.9) | 59.4 (0.00) |
| February | 102(7.4) | 1268(92.6) | | 34(33.3) | 60(58.8) | 8(7.8) | |
| March | 86(6.6) | 1213(93.4) | | 32(37.2) | 52(60.5) | 2(2.3) | |
| April | 118(10.7) | 980(89.3) | | 44(37.3) | 68(57.6) | 8(6.8) | |
| May | 262(16.6) | 312(83.4) | | 100(38.2) | 128(48.8) | 34(12.9) | |
| June | 952(32.2) | 2000(67.8) | | 338(35.5) | 516(54.2) | 98(10.3) | |
| July | 686(28.2) | 1744(71.8) | | 216(31.5) | 378(55.1) | 92(13.4) | |
| August | 307(17.3) | 1466(82.7) | | 83(27) | 202(65.8) | 22(7.2) | |
| September | 330(17.7) | 1536(82.3) | | 110(33.3) | 170(51.5) | 50(15.1) | |
| October | 518(18.3) | 2306(81.7) | | 184(35.5) | 280(54.1) | 54(10.4) | |
| November | 458(16) | 2400(84) | | 151(32.9) | 258(56.3) | 49(10.7) | |
| December | 348(14.4) | 2069(85.6) | | 110(31.6) | 206(59.2) | 32(9.2) | |
| Total | 43,22(17.9) | 19785(82.1) | | 1460(6.1) | 2409(10) | 453(1.9) | Total |

χ²: Pearson chi-square, Mixed: *P. falciparum* and *P.vivax*.

The prevalence of malaria has fluctuated between 2015 and 2023, with some years seeing a rise (e.g., 2019: 19%, 2022: 19.1%) and others a decline (e.g., 2017: 5.6%, 2020: 11%). In 2018 recorded the lowest prevalence of malaria (4%), indicating a substantial reduction in malaria cases that year. However, 2024 stands out with a significant increase in malaria cases (26.9%), which might indicate an outbreak or a change in conditions influencing transmission, such as increased vector population, environmental factors, or reduced preventive measures (Table 2).

The distribution of *Plasmodium* species varied significantly over the ten-year period (Chi-square = 172.1, *P*<0.001). Regarding the proportions of *Plasmodium* infections, *P. falciparum* was the dominant species in 2016 and 2018, accounting for 69.7% and 52.9% of cases, respectively, while *P. vivax* predominated during the other eight years of the study. Additionally, mixed infections (*P. vivax* and *P. falciparum*) showed a significant upward fluctuation, peaking at 20.8% in 2019 and dropping to a minimum of 1.5% in 2018 (Table 2).

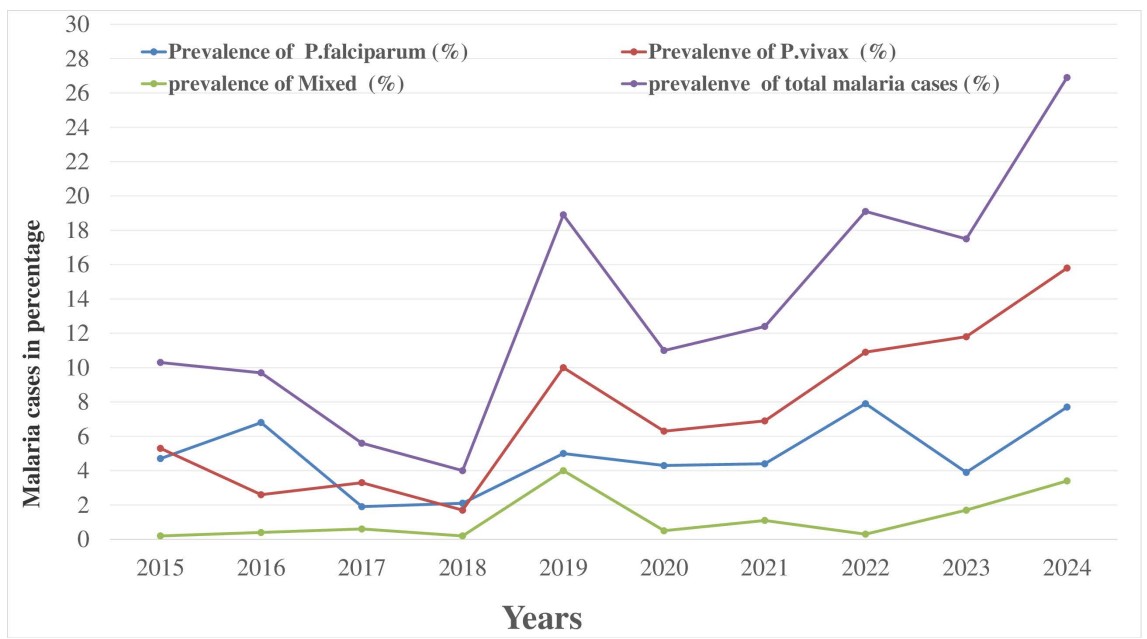

**Fig 1. Annual trend of malaria positivity rates among suspected cases at Bichena Primary Hospital from 2015 to 2024.**

The study showed that *P. vivax* was the dominant species, with notable peaks in 2024 (15.8%), 2023 (11.8%), and 2022 (10.9%), while the lowest prevalence was in 2018 (1.6%). In contrast, *P. falciparum* was the dominant species in 2016 and 2018, with 6.8% and 2.1% of confirmed cases, respectively. Additionally, the prevalence of mixed infections with *P. vivax* and *P. falciparum* showed a fluctuating trend, peaking at 4% in 2019, while minimal cases (0.2%) were observed in 2015 and 2018. These data highlight the dynamics relationship and prevalence between *P. vivax* and *P.falciparum* during the study period, with mixed infections becoming increasingly significant in recent years (Fig 1).

## Seasonal and monthly variations in malaria positivity rates

Malaria prevalence in the region appears to peak during the summer and rainy months (June and July), with the lowest prevalence during the colder winter months (January and February). The highest prevalence of malaria occurs in summer (26.8%), followed by Spring (16.7%).Winter (12.9%) and Autumn (11.7%) show lower malaria prevalence (Table 2). Similarly, the highest prevalence of total confirmed malaria cases was observed in June (32.2%), followed by July (28.2%), while the lowest prevalence occurred in March (6.6%) and February (7.4%) (Table 2). This suggests that malaria transmission is higher during the warmer months, likely due to favorable conditions for the mosquito vectors (temperature and rainfall).

Throughout the 12-month period, *P. vivax* was the most predominant species. The highest prevalence of *P. vivax* and *P. falciparum* was observed in June (17.5% and 11.4% respectively), while the lowest was in March (4% and 2.4% respectively). Mixed infections showed monthly fluctuations, with the highest value in July (3.8%) and the lowest in March (0.2%) (Fig 2).

The prevalence of *P. vivax* infections was consistently dominant across all seasons. The highest *P. vivax* infection rate was recorded in summer (15.1%) (June, July, and August) and the lowest in autumn (6.2%). *P. falciparum* infection was highest in summer (8.9%) and lowest in winter (4.1%). Regarding the proportion of infections with *Plasmodium* species, the highest proportion of *P. vivax* was observed in the winter months (December, January, and February) at 60.6%, and

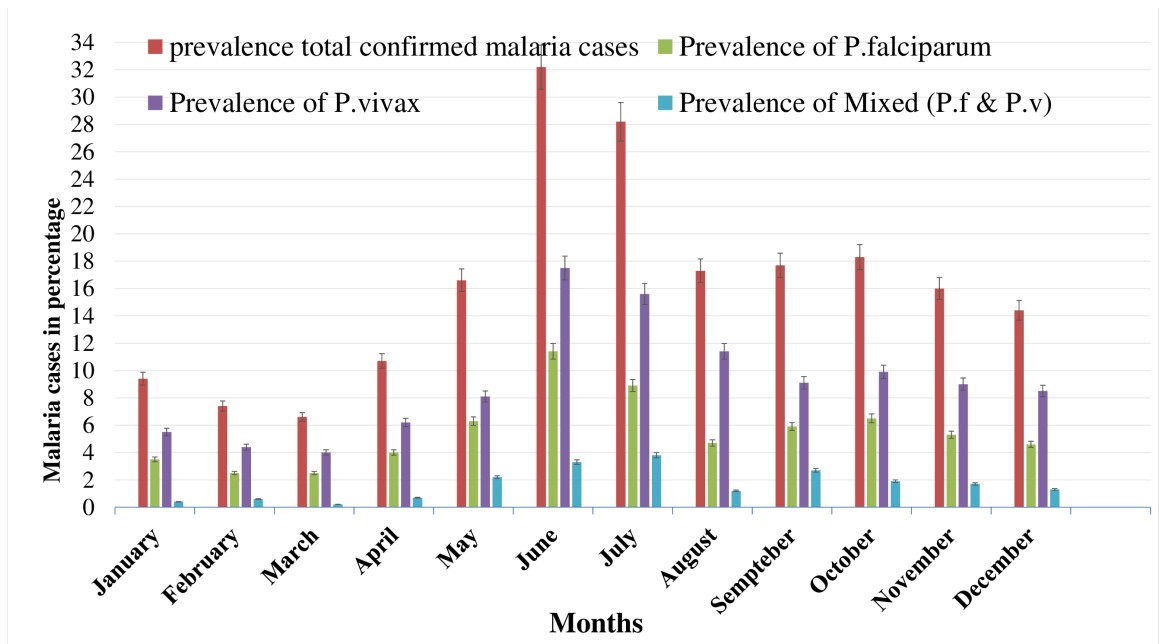

**Fig 2. Monthly prevalence of malaria and proportion of *Plasmodium* species among patients who requested malaria examination at Bichena Primary Hospital from 2015 to 2024.**

the lowest in the spring months (September, October, and November) at 52.2%. In contrast, *P. falciparum* was most common in the autumn months (April, March, and May) at 37.7% and least common in the winter months at 31.4%. Mixed infections were most common in the summer months (June, July, and August) at 11.1% and least common in winter (7.9%) (Table 2 and Fig 3).

## Discussion

The purpose of this trend study was to evaluate the spread and impact of malaria within the catchment area of Bichena Primary Hospital from 2015 to 2024. By identifying key patterns and trends, the analysis aims to inform the priorities of the Ethiopian Malaria Elimination Program and help reduce the disease's impact nationwide.

The prevalence of malaria varies significantly with altitude, as evidenced by multiple studies across different regions. The study found that the malaria prevalence among 24,107 clients, whose blood films were requested and recorded in the laboratory logbooks at Bichena Primary Hospital over a ten-year period, was 17.9% (95% CI: 17.4%-18.4%). This figure is significantly lower than the prevalence rates reported in various areas, such as 60.56% by microscopy and 39.44% by rapid diagnostic tests (RDT) in selected zones of the Amhara region in Northwest Ethiopia [21], 66.7% in Bale Zone, Ethiopia [10], 31.1% in Ziquala District, Northeast Ethiopia [22], 24.7% around Lake Tana and its surrounding areas, Northwest Ethiopia [23], 20.7% at Mizan Tepi University Teaching Hospital, Southwest Ethiopia [24], and 21.8% in Boricha District, Southern Ethiopia [9]. Conversely, several studies have reported lower malaria prevalence rates. For instance, a study at Ataye Hospital in North Shoa, Ethiopia, found a prevalence of just 8.4% over a five-year period [25]. Similarly, another study at Kombolcha Health Centre reported a prevalence of 7.52% over 8 years [26], while 7.7% was observed at the University of Gondar Specialized Referral Hospital [27]. Additionally, 4.5% was found in Southern Ethiopia [28], 4.2% in Mojo Town, Central Ethiopia [29], 15.5% in Northeastern Ethiopia [30], and 13.61% among adults across Ethiopia [31].

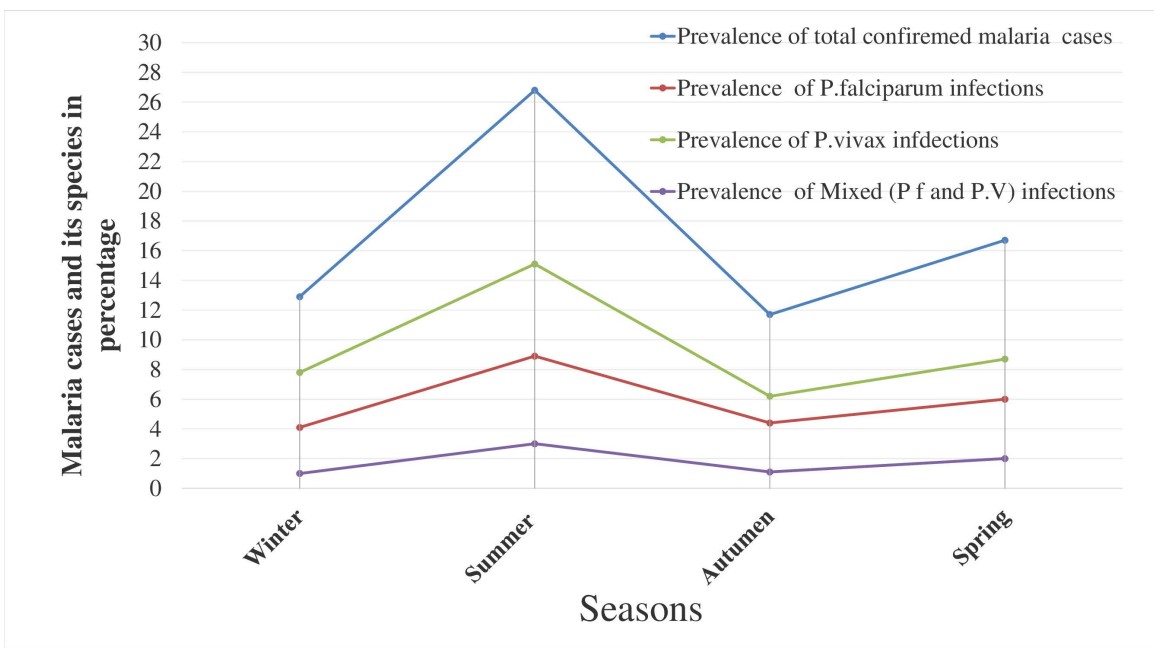

**Fig 3. Seasonal variations in malaria prevalence and the proportion of *Plasmodium* species identified through blood smear microscopy at Bichena Primary Hospital from 2015 to 2024.**

The observed differences in malaria prevalence can be attributed to several factors, such as variations in the quality of malaria diagnosis and the skill level of laboratory personnel in detecting and identifying malaria parasites. Differences in the implementation of malaria prevention and control measures across various regions may also contribute to these disparities. Moreover, demographic factors like sex, age, and geographic conditions, including altitude, temperature, and rainfall, may influence malaria transmission dynamics and prevalence [32]. Additionally, regional variations in economic activities could play a role, as they might affect exposure risks and access to preventive measures. Public awareness regarding the use of insecticide-treated bed nets, knowledge about malaria transmission, and health-seeking behaviors may also differ between populations, further impacting malaria prevalence rates. These factors together suggest that a multifaceted approach is needed to understand and address the variability in malaria prevalence across different areas.

The average annual trend of malaria prevalence indicates that, although there was a decline from 2015 to 2018, a significant increase in malaria cases was observed after 2018, with a particularly sharp rise in the last three study years (2022–2024). This trend highlights a notable uptick in malaria prevalence during this period. A similar report was observed at the University of Gondar Specialized Referral Hospital, Northwest Ethiopia, from 2014 to 2019 [27], and in Boricha District and Bale Zone, Ethiopia, from 2010 to 2017 [9,10]. It is also in line with the recent report by the federal Ministry of Health in Ethiopia [12,13]. The reduction in malaria prevalence during the study period (2015–2018) can be attributed to several factors. These include enhanced malaria control and prevention efforts by various responsible entities, increased community awareness regarding the use of ITNs and IRS, and improvements in mosquito breeding site management through drainage systems. Additionally, climate change may have influenced vector dynamics, affecting transmission patterns. Integrated control strategies, implemented as part of the national malaria control program, likely contributed to these reductions in malaria incidence [27,33]. The subsequent increase in malaria prevalence may be attributed to several factors, including public unrest in the country, insecticide resistance, the COVID-19 pandemic, and the emergence of the

invasive *An. stephensi* vector. These factors likely disrupted ongoing malaria control efforts and contributed to the resurgence of transmission [17,34–36].

Notably, in 2024, there were potential outbreaks or significant changes in malaria transmission patterns. This increase in prevalence may be linked to the political unrest in the region, which resulted in the inability of health extension workers and other essential service providers to operate effectively. Consequently, community-based malaria management, prevention, diagnosis, and treatment were severely neglected. Furthermore, key malaria prevention activities, such as IRS and the distribution of ITNs, were completely halted due to the ongoing political instability and the overall disruption of healthcare services. These factors likely contributed to the escalated malaria transmission observed in 2024.

This study found that *P. vivax* was the predominant species over the ten-year study period, except in 2016 and 2018, although its proportion fluctuated annually. On average, *P. vivax* accounted for 10% of cases, followed by *P. falciparum* at 6.1%, and mixed infections at 1.9%. This finding is consistent with studies conducted in various regions of Ethiopia [8,16,17,27,37]. In contrast, other studies have reported *P. falciparum* as the predominant species [7,9,10,38–40]. The variation in *Plasmodium* species distribution observed in this study may reflect regional differences in factors such as vector species composition, local malaria control measures, and population immunity. The higher prevalence of *P. vivax* in our study could also be associated with the relapse behavior of *P. vivax* cases, which can lead to recurrent infections. These findings highlight the complexity of malaria transmission dynamics, poor implementation of treatment guidelines (radical cure) and underscore the need for region-specific strategies in malaria control [41–43].

Malaria prevalence varied across different seasons, with the highest rate observed during summer (26.8%), decreasing in winter (12.9%) and autumn (11.7%), and these seasonal variations were statistically significant (P<0.001). In contrast, the highest malaria cases in Bale Zone, Ethiopia, were recorded during the spring months (September-November) [10]. These observed seasonal variations in malaria prevalence may be influenced by climate change, which can fluctuate from year to year [44]. During late autumn and early summer in Ethiopia, the rainfall provides favorable conditions for mosquito breeding and malaria transmission. However, later in the summer, most endemic areas often undergo regular indoor residual spraying, which can result in a decline in cases [45]. The two summer months (July and August) are typically characterized by heavy rainfall, which is not generally favorable for mosquito vector propagation. However, variations in rainfall patterns from year to year may alter vector dynamics. Changes in temperature, rainfall, and relative humidity, driven by climate change, are known to directly impact malaria transmission by modifying the behavior and geographic distribution of malaria vectors, as well as by affecting the length of the parasite's life cycle. Additionally, climate change can indirectly influence malaria transmission by altering ecological relationships among the organisms involved, including the vector, parasite, and host [44,45].

The current study revealed that males were more affected by malaria infection than females. The odds of malaria positivity among males were 1.27 times higher than those of females. Similar studies have shown that males are more affected by malaria than females [10,27,46–50]. Conversely, other studies have shown that females are more affected than males [17,46,51]. The difference in sex-based malaria prevalence may be due to various factors, including differences in exposure to mosquito vectors, behavioral patterns, and biological differences between males and females. Males may be more frequently engaged in outdoor activities, increasing their risk of exposure to infected mosquitoes. Additionally, variations in immune response between sexes could contribute to different susceptibility and disease outcomes [52–54]. Another possible reason for the observed difference in malaria prevalence could be that males are more mobile, traveling to malaria-endemic areas in search of temporary employment. In contrast, females are less likely to engage in outdoor activities and typically stay at home, often taking on roles such as cooking or household duties. This reduced mobility and outdoor exposure among females may lower their risk of malaria infection compared to males.

Age was also a contributing factor to the prevalence of malaria, with the highest rates observed in the 15–24 age group compared to older age groups. The odds of malaria positivity among individuals in the 15–24, 25–34, and 35–44

age groups were 1.27, 1.01, and 1.02 times higher, respectively, than those in the age group older than 45 years. Similarly, other studies report the highest malaria infection in age group 15–24 [7–9,27,55–57]. However, in this study, children under five years old and those in the 5–14 age group had odds of malaria positivity that were 0.42 and 0.84 times lower, respectively, compared to the older age group (≥45 years) and it is in contrast to other studies [10,27,55–58]. The differences in malaria prevalence across age groups can be attributed to several factors. Higher rates in the 15–44 age groups may result from increased exposure due to outdoor activities, agricultural work, and mobility, as individuals in these groups are more active and spend time in malaria-endemic areas. Additionally, long-distance travel for schooling increases their risk. In contrast, lower prevalence in children under five could be due to maternal immunity, while children aged 5–14 may have developed partial immunity through repeated exposures. Those over 45 may have acquired stronger immunity due to long-term exposure. These factors reflect the complex relationship between age, immunity, and exposure in malaria risk [8,59,60]. This variability underscores the importance of local epidemiological data in guiding malaria management strategies.

The study possessed several strengths. First, it had a sufficiently large sample size, which enhanced its statistical power. Second, it included participants of all age groups, from children to older adults, providing a comprehensive representation of the population. However, there were some limitations. The data relied on laboratory logbooks, which omitted important variables like participants' body temperature, clinical presentations, or residential information. Additionally, the study lacked detailed data on major interventions against malaria, and key environmental factors were neither collected nor considered. As a secondary data analysis, the reliability of the recorded data could not be fully verified.

## Conclusion

The current findings indicate a significant upward trend in malaria prevalence in the study area, particularly after 2018, making malaria a major public health concern that necessitates intensified efforts for further reduction. On average, the highest number of malaria cases was observed during the summer months. However, the prevalence exhibited considerable fluctuation from year to year, with a notably higher peak in 2024. Younger age groups, particularly those aged 15–24, were more affected compared to older age groups and children under five. While *P. vivax* was the predominant *Plasmodium* species overall, there was considerable variation in the prevalence of *P. falciparum* and *P.vivax* from year to year and across seasons. Given these fluctuations, it is essential to continue and strengthen prevention and control measures in the study area, taking into account these temporal and demographic variabilities.

## Supporting information

**S1 File. Data files of this study finding in SPSS software.**
(SAV)

**S2 File. Excel data used for developing Fig 1, 2, and 3.**
(XLS)

## Acknowledgments

We would like to thank the laboratory staff of Bichena Primary Hospital for their contributions to data collection. We also extend our gratitude to Debre Markos University and Bichena Primary Hospital for granting permission to conduct this research and PhD study.

## Author contributions

**Conceptualization:** Awoke Minwuyelet, Delenasaw Yewhalaw, Getnet Atenafu.
**Data curation:** Awoke Minwuyelet.

**Formal analysis:** Awoke Minwuyelet.

**Investigation:** Awoke Minwuyelet.

**Methodology:** Awoke Minwuyelet.

**Resources:** Awoke Minwuyelet.

**Software:** Awoke Minwuyelet.

**Supervision:** Delenasaw Yewhalaw, Getnet Atenafu.

**Visualization:** Awoke Minwuyelet.

**Writing – original draft:** Awoke Minwuyelet.

**Writing – review & editing:** Awoke Minwuyelet, Delenasaw Yewhalaw, Getnet Atenafu.

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
