## [Decision Letter · Decision Letter 0]

21 Feb 2025

PONE-D-25-03723Retrospective Analysis of Malaria Prevalence Over Ten Years (2015-2024) at Bichena Primary Hospital, Amhara Region, EthiopiaPLOS ONE

Dear Dr. Minwuyelet,

Thank you for submitting your manuscript to PLOS ONE. After careful consideration, we feel that it has merit but does not fully meet PLOS ONE’s publication criteria as it currently stands. Therefore, we invite you to submit a revised version of the manuscript that addresses the points raised during the review process. Kindly see reviwer comments, especially those of reviewer 1 below.

We look forward to receiving your revised manuscript.

Kind regards,

Benedikt Ley, PhD

Academic Editor

PLOS ONE

Journal Requirements:

Reviewers' comments:

Reviewer's Responses to Questions

**Comments to the Author**

1. Is the manuscript technically sound, and do the data support the conclusions?

Reviewer #1: Yes

Reviewer #2: Yes

2. Has the statistical analysis been performed appropriately and rigorously?

Reviewer #1: Yes

Reviewer #2: Yes

3. Have the authors made all data underlying the findings in their manuscript fully available?

Reviewer #1: No

Reviewer #2: Yes

4. Is the manuscript presented in an intelligible fashion and written in standard English?

Reviewer #1: Yes

Reviewer #2: No

5. Review Comments to the Author

Reviewer #1: Discretionary Revisions:

1. Please include a description of the univariate logistic analysis in the data analysis section. Also need to added what type of categorical test was used (pearson / spearman)

2. It is necessary to revise Table 1 and amend the notation: the age category 514 should be corrected to 5-14, the total percentage inside the age category must equal 100, the p-value for the COR should be included, and the reference values should be presented consistently, either all in the first line or all in the final line.

3. What are the reasons that more than 40% blood films were examine in 2024?

4. What are the justifications for segregating the data for the periods 2015-2017 and 2018-2024, whereas for malaria species, the periods are 2015-2018 and 2019-2024?

5. Table 2 requires revision due to improper presentation of the percentages. Need ot remove the .00 after the years.

6. The presentation of the chi-square and p-value in this manner is misleading and fosters misunderstanding. It is preferable to give, for example, χ2 (p-value), with the p-value displayed to either two or three decimal places.

7. The Y-axis of figures should be labeled as malaria cases expressed in percentage.

8. Are all four seasons present in the study area? If not, please provide a reference for the seasons analyzed in the study.

9. What are the reasons that the prevalence is high in summer but the detection rate is high in winter for P.vivax and similar pattern exist for P.falciparum i.e. prevalence high in summer and detection rate is high in autumn?

Minor Essential Revisions

Line # 145: data was collected by the first author and other trained laboratory personnel using a data extraction sheet established for this study – What does the term "extraction sheet" mean? Do you enter data in an Excel file or a paper form? What steps were taken to prevent data entry errors?

Line # 159: At least two laboratory technologists before being reported. In cases of discordant results, a third senior technician reexamined the slide - is this the standard procedure at the hospital, it seems quite strange in this environment. If not how 24143 slides were evaluated within 16 days?

Line # 195: More than half of the patients 12,973 (53.4%) were male, - in table-1 it is 12,873, please take corrective measure.

Line # 207: monthly, seasonal, and yearly patterns – in table-1, there are not information regarding monthly or yearly patterns, please remove it.

Line # 240: Additionally, the prevalence of mixed infections with P. vivax and P. falciparum exhibited a clear upward trend, peaking at 4% in 2019, whereas only minimal cases (0.2%) were noted in 2015 and 2018 – The trend is not upward clearly, it is ups and down thus please revised the sentence properly.

Line # 342-345: Repeated sentence please remove one.

Reviewer #2: Dear Author, This a great work. Please see the following few comments and questions.

Line 81; what progress has been seen in the past two decades? Please elaborate on it.

Line 120 and 121: can you specifically mention when the outbreak happened?

Lines 151-161: has to go somewhere or need a separate subtitle

Line 158-161: Is the number of slide readers the standard across Ethiopia or the hospital? It looks unrealistic for a blood film to be read by two technologists, even though there are more technicians than technologists in most health facilities. It may never come across a slide to be read by a third reader.

Line 203: change the title to only “Malaria distribution.”

Line 204-207: This sentence came a bit earlier than the following results. Rewrite and move down.

Lines 237-241 and Lines 253-258 look to oppose each other.

Line 338 -339: The MoH report is nationwide and can not be compared with a local report.

Line 346: not only “complexity of malaria transmission dynamics” but also poor implementation of treatment guidelines(radical cure)

Line 413: Why only after 2021? From the findings, it looks from 2018.

Line 420: The sentence “different Plasmodium species “ looks that the study revealed other plasmodium species other than vivax and falciparum.

General comments

The last date of data collection is January 14/2025. It looks unrealistic to do data analysis and draft a manuscript to send for publication.

Please explain how data quality and completeness are maintained.

Would be great to share the ethical approval letter.

Results: good to start with demographics first, followed by a description of malaria

Table 1 is congested.

In the discussion, comparing the prevalence based on the area's similarity in altitude and other characters is good.

The second age category in Table 1 needs correction.

6. PLOS authors have the option to publish the peer review history of their article (what does this mean? ). If published, this will include your full peer review and any attached files.

**Do you want your identity to be public for this peer review?** For information about this choice, including consent withdrawal, please see our Privacy Policy .

Reviewer #1: No

Reviewer #2: **Yes: ** Tamiru Shibiru Degaga

---

## [Author Response · Author response to Decision Letter 1]

26 Feb 2025

Dear Editors and Reviewers,

First of all, we would like to thank you and the anonymous reviewers for the valuable and constructive comments on our manuscript titled "Retrospective Analysis of Malaria Prevalence Over Ten Years (2015-2024) at Bichena Primary Hospital, Amhara Region, Ethiopia: manuscript ID: PONE-D-25-03723 -EMID:9e088c3dadc53452. We have addressed all the comments given point by point and revised the manuscript accordingly. The responses to all the comments are incorporated in the revised manuscript. Below, please find the point by point responses to the comments:

Best regards!

Awoke Minwuyelet, Corresponding author.

Here is a point by point response to the reviewers’ and editor comments and concerns:

Responses for Reviewer's and editor Responses to Questions

Comments #1. Is the manuscript technically sound, and do the data support the conclusions?

Reviewer #1: Yes

Reviewer #2: Yes

Response#1. Thank you for your positive assessment of the technical soundness of our manuscript. We greatly appreciate your time and effort in reviewing our work. Your feedback is invaluable, and we are pleased that you find our study rigorous and well-supported by the data.

Comment #2. Has the statistical analysis been performed appropriately and rigorously?

Reviewer #1: Yes

Reviewer #2: Yes

Response #2. Thank you for your positive evaluation of our statistical analysis. We appreciate your time and thoughtful review, and we are glad that you find our approach appropriate and rigorous.

Comment #3. Have the authors made all data underlying the findings in their manuscript fully available?

Reviewer #1: No

Reviewer #2: Yes

Response #3. We sincerely appreciate your recommendations and support. In response to your suggestion, we included the data availability in the form of a supplementary file with the revised manuscript. However, we have believed that the individual data points behind means, medians, and variance measures may not add significant value to the manuscript. Instead, we have presented the data using appropriate statistical analyses, including multivariable regression and the chi-square test, along with relevant figures. Additionally, all the data and figures were developed by the primary author of the manuscript, and no figures have been used from any other sources for analysis.

Comment #4. Is the manuscript presented in an intelligible fashion and written in standard English?

Reviewer #1: Yes

Reviewer #2: No

Response #4. We also accept the recommendation and have addressed the language mistakes throughout the revised manuscript.

Responses for Reviewer #1: Reviewer’s Report

Comment #1. Please include a description of the univariate logistic analysis in the data analysis section. Also need to added what type of categorical test was used (pearson / spearman).

Respnse#1. We appreciate your feedback and have made the necessary corrections in the revised version of the manuscript as per your recommendation.

Comment #2. It is necessary to revise Table 1 and amend the notation: the age category 514 should be corrected to 5-14, the total percentage inside the age category must equal 100, the p-value for the COR should be included, and the reference values should be presented consistently, either all in the first line or all in the final line.

Response #2. We have corrected the age category to 5–14 in Table 1. However, the recommendation to include the p-value for the COR and present reference values consistently in multivariable regression tables is significant for enhancing clarity and interpretability. This practice addresses common reporting issues that can lead to misinterpretation of statistical results. The reference values can also be presented based on the study's objectives and the author's intended description of the findings compare to other findings. Additionally, the P-value of the COR is not emphasized in this table, as the confidence interval (CI) of the odds ratio provides a more meaningful interpretation of statistical significance and presenting the P-value makes the table more congested. Different findings were also presented in same manner (Plasmodium gametocyte carriage in humans and sporozoite rate in anopheline mosquitoes in Gondar zuria district, Northwest Ethiopia. Prevalence of malaria and associated factors among children attending health institutions at South Gondar Zone, Northwest Ethiopia: a cross-sectional study).

Comment #3. What are the reasons that more than 40% blood films were examine in 2024?

Respnse#3. Thank you for this important question. In the region, there has been ongoing public unrest and continuous conflict since July 2023, which has severely impacted healthcare services. As a result, health extension workers and other essential service providers have been unable to operate, leading to the neglect of community-based malaria management, prevention, diagnosis, and treatment. Additionally, critical malaria prevention activities, such as indoor residual spraying (IRS) and insecticide-treated net (ITN) distribution, have been completely halted due to political instability and the disruption of healthcare services. However, hospitals have maintained relatively better service availability and drug accessibility, which may have contributed to the increased number of reported malaria cases. Furthermore, a malaria outbreak occurred during the year, further exacerbating the situation.

Comment #4. What are the justifications for segregating the data for the periods 2015-2017 and 2018-2024, whereas for malaria species, the periods are 2015-2018 and 2019-2024?

Response #4. Sorry for any confusion. Rather than segregating by years, our revision separates the Chi-square ratio and its P-value more clearly. As per your request, we have made the necessary modifications in the revised version (Table 2).

Comment#5. Table 2 requires revision due to improper presentation of the percentages. Need ot remove the .00 after the years.

Response #5. Thank you for your recommendation. We have made the necessary corrections accordingly (Table 2).

Comment #6. The presentation of the chi-square and p-value in this manner is misleading and fosters misunderstanding. It is preferable to give, for example, χ2 (p-value), with the p-value displayed to either two or three decimal places.

Response#6: Thank you for your recommendation. We have made the necessary corrections accordingly (Table 2).

Comment #7. The Y-axis of figures should be labeled as malaria cases expressed in percentage.

Response #7. We corrected as per your recommendations (fig 1, 2 and 3)

Comment #8. Are all four seasons present in the study area? If not, please provide a reference for the seasons analyzed in the study.

Response #8. Yes. The Amhara region of Ethiopia experiences distinct seasonal variations, although not all four seasons are equally pronounced. The primary seasons identified in the studies include summer, autumn, winter, and spring with significant climatic events affecting agricultural practices and malaria incidence. Summer (June to September)( Analysis of Long-Term Trends of Annual and Seasonal Rainfall in the Awash River Basin, Ethiopia: Autumn (October to January)( Climatic risk adaptation strategies by smallholder livestock farmers in Eastern Amhara Region, Ethiopia), Winter (February to May)( Analysis of Long-Term Trends of Annual and Seasonal Rainfall in the Awash River Basin, Ethiopia) and Spring (March, to May)( Innovative Trend Analysis of Annual and Seasonal Rainfall Variability in Amhara Regional State, Ethiopia.

Comment #9. What are the reasons that the prevalence is high in summer but the detection rate is high in winter for P.vivax and similar pattern exist for P.falciparum i.e. prevalence high in summer and detection rate is high in autumn?

Response #9. Thank you for raising this important question. In this region, the months of May and June typically experience rainfall that creates favorable conditions for vector breeding, which in turn exacerbates malaria transmission. During this period, communities are often preoccupied with agricultural activities, leading to a neglect of malaria prevention measures. However, every year, early July marks the start of indoor residual spraying (IRS) in endemic areas, along with the distribution of insecticide-treated nets (ITNs). Despite these efforts, cases often continue to occur due to pre-existing infections from the summer season (June to September). Interestingly, while the summer season is marked by heavy rainfall, which typically washes away breeding sites, the autumn months (October to January) also provide favorable conditions for vector breeding and malaria transmission. Though IRS and ITN application in preceding months may reduce malaria cases, improper or insufficient application of these interventions can still result in a high infection rate during the later transmission periods.

Minor Essential Revisions

Comment #10. Line # 145: data was collected by the first author and other trained laboratory personnel using a data extraction sheet established for this study – What does the term "extraction sheet" mean? Do you enter data in an Excel file or a paper form? What steps were taken to prevent data entry errors?

Response #10.The data was first collected using paper form extraction sheet from lab registration book then entered to Soft war. To facilitate ease of analysis and to reduce manual errors associated with paper forms and prevent data entry errors, the following measures were implemented;

1. Training of Personnel: All personnel involved in data collection and entry were trained to use the data extraction sheet effectively and consistently, minimizing variability in data recording.

2. Data Validation Checks: Validation rules were applied in paper form to ensure that data entered into each field met predefined criteria (like: correct data types, within expected ranges).

3. Regular Audits: Periodic checks and audits were performed to verify the accuracy of the data entries, and corrections were made where necessary.

4. Clear Definitions: Detailed definitions and instructions for each variable on the extraction sheet ensured clarity for data collectors, reducing the risk of misinterpretation.

5. Finally, the paper for collected data was entered to SPSS for analysis. These steps, combined with regular review and feedback, helped ensure the accuracy and integrity of the data collected for this study.

Comment #11. Line # 159: At least two laboratory technologists before being reported. In cases of discordant results, a third senior technician reexamined the slide - is this the standard procedure at the hospital, it seems quite strange in this environment. If not how 24143 slides were evaluated within 16 days?

Response #11. Thank you for raising this important point. At this hospital, the primary author, who has served as the head of the laboratory department since its inception, oversaw the development of standardized protocols for the review of various diagnostic techniques. These protocols included procedures for the rereading of blood film staining, Zehlenhson AFB, staining, fluorescent AFB microscopy, and Gram staining microscopy. In the event of any discordance between two primary readers, a third reader was involved to mediate and ensure consensus, thereby ensuring the accuracy and reliability of the final results before they were reported.

Please note that this was a retrospective study, in which data was collected from 24,143 participants documented in the registration book at the time of the study. Microscopic examination was not performed during the data collection phase. However, the diagnostic approach was based on the aforementioned principles, ensuring consistency and adherence to established protocols for accurate diagnosis, including blood film staining, Zehlenhson staining, fluorescent AFB microscopy, and Gram staining microscopy, as outlined previously.

Comment #12. Line # 195: More than half of the patients 12,973 (53.4%) were male, - in table-1 it is 12,873, please take corrective measure.

Response #12. We accept your feedback and have made the necessary corrections based on your recommendation (Line 206).

Comment #13. Line # 207: monthly, seasonal, and yearly patterns – in table-1, there are not information regarding monthly or yearly patterns, please remove it.

Response #13. Thank you for pointing this out. Considering your comment, we have added the information to the manuscript rather than removing it, as it is crucial for providing context to the observed changes in malaria prevalence (line 242-248 and 272-280).

Comment #14. Line # 240: Additionally, the prevalence of mixed infections with P. vivax and P. falciparum exhibited a clear upward trend, peaking at 4% in 2019, whereas only minimal cases (0.2%) were noted in 2015 and 2018 – The trend is not upward clearly, it is ups and down thus please revised the sentence properly.

Response #14. We appreciate your insights and we have made the corrections as per your recommendations (Line: 260-262).

Comment #15. Line # 342-345: Repeated sentence please remove one.

Response #15. We accept your recommendations and have made the necessary corrections accordingly( Line 379).

Responses for Reviewer #2: Reviewer’s Report

Comment #1. Line 81; what progress has been seen in the past two decades? Please elaborate on it.

Response #1. Thank you, for your questions. Malaria remains a significant public health and socioeconomic challenge in Ethiopia, despite substantial progress over the past two decades. Efforts such as widespread distribution of insecticide-treated nets (ITNs), indoor residual spraying (IRS), and improvements in diagnostic and treatment services have contributed to a decline in malaria-related morbidity and mortality in some regions. However, persistent challenges such as geographic variability, seasonal rainfall, and vector resistance to insecticides continue to hinder control efforts. Additionally, socioeconomic factors like limited access to healthcare in rural areas, poverty, and political instability further exacerbate the situation, impeding the effectiveness of prevention and treatment programs. The economic burden of malaria, including lost productivity and healthcare costs, remains high, and displacement due to conflict or migration can lead to further spread of the disease. Thus, while progress has been made, malaria continues to affect the health and economy of Ethiopia, requiring sustained and multifaceted interventions ( line 78-88).

Comment #2. Line 120 and 121: can you specifically mention when the outbreak happened?

Response #2. While malaria incidence has been steadily increasing since 2022 in the region, the malaria outbreak was specifically registered in starting June 2024 up November 30/2024.

Comment #3. Lines 151-161: has to go somewhere or need a separate subtitle

Response #3. We added subtitles based on the recommendations (Line 160).

Comment #4. Line 158-161: Is the number of slide readers the standard across Ethiopia or the hospital? It looks unrealistic for a blood film to be read by two

---

## [Decision Letter · Decision Letter 1]

24 Mar 2025

Retrospective Analysis of Malaria Prevalence Over Ten Years (2015-2024) at Bichena Primary Hospital, Amhara Region, Ethiopia

PONE-D-25-03723R1

Dear Dr. Minwuyelet,

We’re pleased to inform you that your manuscript has been judged scientifically suitable for publication and will be formally accepted for publication once it meets all outstanding technical requirements.

Kind regards,

Benedikt Ley, PhD

Academic Editor

PLOS ONE

Additional Editor Comments (optional):

Reviewers' comments:

Reviewer's Responses to Questions

**Comments to the Author**

1. If the authors have adequately addressed your comments raised in a previous round of review and you feel that this manuscript is now acceptable for publication, you may indicate that here to bypass the “Comments to the Author” section, enter your conflict of interest statement in the “Confidential to Editor” section, and submit your "Accept" recommendation.

Reviewer #1: All comments have been addressed

2. Is the manuscript technically sound, and do the data support the conclusions?

Reviewer #1: Yes

3. Has the statistical analysis been performed appropriately and rigorously?

Reviewer #1: Yes

4. Have the authors made all data underlying the findings in their manuscript fully available?

Reviewer #1: Yes

5. Is the manuscript presented in an intelligible fashion and written in standard English?

Reviewer #1: Yes

6. Review Comments to the Author

Reviewer #1: Authors have reviewed the responses, and each one meets the required standards. I have no additional comments on this matter.

7. PLOS authors have the option to publish the peer review history of their article (what does this mean? ). If published, this will include your full peer review and any attached files.

**Do you want your identity to be public for this peer review?** For information about this choice, including consent withdrawal, please see our Privacy Policy .

Reviewer #1: No

---

## [Editor Report · Acceptance letter]

PONE-D-25-03723R1

PLOS ONE

Dear Dr. Minwuyelet,

I'm pleased to inform you that your manuscript has been deemed suitable for publication in PLOS ONE. Congratulations! Your manuscript is now being handed over to our production team.

Kind regards,

on behalf of

Dr Benedikt Ley

Academic Editor

PLOS ONE